# Skin Cancer Correlations in Psoriatic Patients

**DOI:** 10.3390/cancers15092451

**Published:** 2023-04-25

**Authors:** Daniel Octavian Costache, Horia Bejan, Marcela Poenaru, Raluca Simona Costache

**Affiliations:** 1II Dermatology Discipline, Faculty of Medicine, Carol Davila University of Medicine and Pharmacy, 050474 Bucharest, Romania; daniel.costache@umfcd.ro; 2Faculty of Medicine, Carol Davila University of Medicine and Pharmacy, 050474 Bucharest, Romania; 3Dermatology Department, Carol Davila University Central Emergency Military Hospital, 010825 Bucharest, Romania; marcelapoenaru@yahoo.com; 4Internal Medicine and Gastroenterology Discipline, Faculty of Medicine, Carol Davila University of Medicine and Pharmacy, 050474 Bucharest, Romania

**Keywords:** psoriasis, skin cancer, treatment

## Abstract

**Simple Summary:**

Psoriasis is an inflammatory disease associated with important comorbidities that shape a specific clinical frame. Concerns have been expressed in recent decades about the role of systemic inflammation in the promotion of various organ neoplasia. Moreover, psoriasis treatment is chronic, usually lifelong, and uses various classes of drugs, of which some have an intrinsic effect on the body’s cancer-control mechanisms. While in other works we studied the correlation between psoriasis, its treatment, and organ cancers, this paper is a review of available data that evaluate the connection and intricate risk of skin cancer in psoriatic patients. Good knowledge of these risks might help both the patient and the care provider to avoid the development of skin neoplasia in selected patients.

**Abstract:**

Psoriasis is a common chronic, immune-mediated, inflammatory disease with associated comorbidities. Common psoriasis-associated comorbidities include psoriatic arthritis, cardiovascular disease, metabolic syndrome, inflammatory digestive syndromes, and depression. A less studied association is between psoriasis and specific-site cancers. A key cell in the pathophysiology of psoriasis is the myeloid dendritic cell, which links the innate and adaptive immune systems, and therefore is involved in the control of cancer-prevention mechanisms. The relationship between cancer and inflammation is not new, with inflammation being recognized as a key element in the development of neoplastic foci. Infection leads to the development of local chronic inflammation, which further leads to the accumulation of inflammatory cells. Various phagocytes produce reactive oxygen species that cause mutations in cellular DNA and lead to the perpetuation of cells with altered genomes. Therefore, in inflammatory sites, there will be a multiplication of cells with damaged DNA, leading to tumor cells. Over the years, scientists have tried to assess the extent to which psoriasis can increase the risk of developing skin cancer. Our aim is to review the available data and present some information that might help both the patients and the care providers in properly managing psoriatic patients to prevent skin cancer development.

## 1. Introduction

Psoriasis is a common chronic, immune-mediated, inflammatory disease characterized by erythematous, scaly plaques that often appear on the elbows, scalp, knees, and lower back, but can affect any surface of the skin [1]. The prevalence of the disease varies according to geographical region and ethnic groups, with the lowest prevalence, 0.17%, in eastern Asia and the highest, 8%, in the Scandinavian peninsula [2].

Due to recent medical discoveries, the paradigm that psoriasis is a limited skin condition has been changed and systemic involvement is now proven, especially in severe forms. Furthermore, the presence of associated comorbidities with a major negative impact on the patient’s quality of life is nowadays a fact. Common psoriasis-associated comorbidities include psoriatic arthritis, cardiovascular disease, metabolic syndrome, diabetes mellitus, inflammatory digestive syndromes, and depression [3,4].

A less studied association is between psoriasis and the risk of developing specific site cancers. Considering that in psoriasis the patient has chronic systemic inflammation and that severe forms of psoriasis are treated with immunosuppressive medications, the hypothesis that psoriasis patients might have a higher risk of developing various forms of cancer can be discussed [5]. Moreover, the percentage of patients with severe psoriasis who smoke is higher than in the general population; the same trend can be observed for alcohol consumption, which might also add to the overall cancer risk [6].

Recent studies have focused on assessing psoriasis patients’ risk of developing skin cancer; however, the results of the studies are heterogeneous, and with the increasing availability of biological therapies, the question of an additional increase in skin cancer risk has been raised [7].

This review aims to better understand the mechanisms by which psoriasis itself can lead to skin cancer and whether systemic therapy administered to patients with psoriasis can influence the risk of skin cancer.

## 2. Pathogenesis of Psoriasis

The specific etiology of psoriasis is not completely understood. However, recent data have shown that genetic susceptibility, environmental factors, and immune dysfunction are all necessary for the disease to develop [8].

Psoriasis is characterized by three major histological features: epidermal hyperplasia (acanthosis), dilated blood vessels in the dermis, and inflammatory infiltration of leucocytes, especially in the dermis. The skin regions not affected by psoriasis plaques show normal histology. Under expression of keratinocyte differentiation markers, a loss of the granular cell layer, parakeratosis (retention of nuclei in stratum corneum cells), the elongation of rete ridges, and the presence of micro-pustules of Kogoj and micro-abscesses of Munro are all associated with hyperplastic epidermal changes [9]. In addition to the above-mentioned changes, the dermis is rich in dendritic cell, macrophage, neutrophil, and CD4+ and CD8+ T-cell inflammatory infiltrates. This infiltration seems to appear before epidermal hyperplasia.

Psoriasis can be classified as a predominantly Th1 response disorder, with increased concentrations of IL-2, IL-12, and interferon-γ. In addition to those cytokines, IL-1β, IL-13, IL-17A, IL-22, IL-23, and TNF-α are also found in psoriatic plaques in greater concentrations [10].

A key cell in the pathophysiology of psoriasis is the myeloid dendritic cell, which links the innate and adaptive immune systems. Histologically, it has been demonstrated that there is an increased number of activated dendritic cells in psoriatic lesions. Furthermore, the dendritic cell is the main interferon-α producer, a cytokine shown to be responsible for the induction of psoriasis [11,12]. An essential step during the development of psoriasis is the transition of T cells from the superficial dermis into the epidermis. This is achieved through the interaction of type IV collagen, from the basement membrane of the epidermis, with T cells, an event facilitated by α1β1 integrin. Experimental clinical studies have shown that blocking the T-cell transition can stop the onset of psoriasis [13].

Due to all these modifications, the disease is associated with the increased production of reactive oxygen species. Under these conditions of persistent inflammation and epidermal hyperplasia, the formation of skin neoplasms may be facilitated [14].

## 3. Concerns Regarding Skin Cancer

The relationship between cancer and inflammation is not new. Inflammation is a key element in the development of neoplastic foci. Infection sites lead to the development of local chronic inflammation, which further leads to the accumulation of inflammatory cells. Various phagocytes produce reactive oxygen species that cause mutations in cellular DNA and lead to the perpetuation of cells with altered genomes. Therefore, in the inflammatory site, there will be a multiplication of cells with damaged DNA, leading to tumor cells [15]. Second, alterations in tumor suppressor genes have been found. The p53 protein plays a major role in tumor suppression by regulating cell division. In patients with chronic inflammation, the gene that produces the p53 protein was found to have multiple mutations, resulting in uncontrolled cell division [16,17]. This process is repetitive, long-lasting, and sustains the multiplication of tumor cells, which eventually leads to the development of the neoplasm [15,18].

Examples supporting the causal relationship between inflammation and cancer include an increased risk of colon cancer in patients with Crohn’s disease or ulcerative colitis and an increased risk of gastric cancer due to infection with Helicobacter pylori [19,20]. With the above hypothesis in mind, it is justified to ask whether psoriasis puts the patient at special risk for the development of neoplasms.

Skin cancer is one of the most studied cancers that psoriasis might cause. Skin cancer can be divided into NMSC (non-melanoma skin cancer) and melanoma skin cancer. In turn, basal cell carcinoma and squamous cell carcinoma are the most frequently encountered NMSCs [21].

Over the years, scientists have tried to assess the extent to which psoriasis can increase the risk of developing skin cancer. The strongest association seems to be between psoriasis and squamous cell carcinoma. A study conducted in the USA, in which the participants were exclusively women, supports the above statement [22]. According to those data, psoriasis patients had a quantifiable higher risk of developing squamous cell carcinoma and melanoma. For basal cell carcinoma, no causal relationship with psoriasis was demonstrated. The risk of developing squamous cell carcinoma was higher in patients under systemic medication, therefore with severe forms of psoriasis [22]. Margolis et al. studied the skin cancer rates in patients with psoriasis and patients with hypertension. The study concluded that patients with severe forms of psoriasis had a higher risk of developing lymphoma or NMSC compared to patients with hypertension. This may be partly explained by the fact that patients with severe forms of psoriasis receive systemic therapies, such as PUVA or Cyclosporine, which increase the risk of developing cancer [23].

However, the risk of developing NMSC appears to be independent of psoriasis severity. A 15-year study in Denmark involving over 5 million participants found that patients with mild forms, not requiring systemic therapy, have a slightly increased risk of developing NMSC and melanoma. Another interesting finding observed in this study is that severe-form psoriasis patients have an increased risk of developing NMSC and not melanoma. The risk of developing NMSC was higher in patients with severe psoriasis [24].

A study conducted in Italy on a sample of 72,000 people comes up with an interesting hypothesis that psoriasis may protect patients from skin cancer. Psoriasis patients showed a relatively lower risk than the control cohort, regardless of the variables considered. However, psoriasis patients treated with PUVA had a higher risk of developing NMSC than those who had never received PUVA. Moreover, the risk of developing NMSC appeared to correlate with the number of cycles of PUVA performed [25].

When discussing psoriasis and the risk of developing melanoma, the scientific evidence is quite mixed. Some studies claim that there is a higher risk of developing melanoma [26,27], while other studies have found no causal relationship between melanoma and psoriasis [28,29]. Moreover, a study in Italy claims that psoriasis patients are protected from melanoma and have a lower risk than the general population [30]. Of the systemic treatments used for patients with severe forms of psoriasis, PUVA appears to increase the risk of melanoma, especially among patients who have had more than 250 cycles of therapy. It is interesting to mention that this result was only seen in the American PUVA cohort study, while the two European PUVA cohorts found no increased risk [31]. In a case–control study conducted in Sweden, methotrexate use was not associated with an increased risk of melanoma [32]. A study published in 2020 claims that patients receiving biologic therapy are not at increased risk of developing melanoma compared to those receiving conventional systemic therapy [33]. Another study from Sweden claims that patients with psoriasis have an increased risk of developing melanoma, with the association being stronger for melanoma in situ [34].

Furthermore, psoriasis is reported to be associated with an increased risk of developing lymphomas. Lymphomas can be divided into Hodgkin’s lymphomas and non-Hodgkin’s lymphomas, which are also the most common. Non-Hodgkin’s lymphomas, in turn, can be divided into B-lymphocyte-origin or T-lymphocyte-origin lymphomas. The most common form of non-Hodgkin’s T-lymphocyte-origin lymphoma is cutaneous T-cell lymphoma (CTCL) [35]. Because of the relatively low number of cases, studying the association between psoriasis and lymphoma is quite difficult; however, it has been pointed out that patients with psoriasis are associated with an increased risk of developing Hodgkin’s lymphomas and CTCL, especially among patients with severe forms of psoriasis who are under systemic treatment [35,36]. Methotrexate and PUVA are the systemic therapies considered to increase the risk of lymphoma in severe psoriasis patients [37]. An English study has shown that psoriasis patients over the age of 65 have a three times higher risk of developing lymphomas than the general population [38].

The increased risk can be explained by a variety of factors, including the patient’s chronic inflammatory state, the disease’s severity, the various systemic treatments that patients with severe forms take, and last but not least the patient’s harmful behaviors, including smoking and alcohol consumption [7].

As mentioned above, psoriasis is characterized by chronic inflammation and epidermal hyperproliferation, both of which can lead to the development of tumor cell clones that eventually lead to cancer [10]. Patients with more severe forms of psoriasis have a higher incidence of cancers, including skin cancers and lymphomas. This may be explained by the fact that patients with more severe forms of psoriasis are undergoing systemic therapies and/or PUVA treatments [39], but also because of the more intense systemic inflammation.

Due to the impaired quality of life, psoriasis patients, especially those with severe forms, are prone to risk factors such as smoking and alcohol consumption [40].

Studies in Italy [41], Finland [42], Norway [43], the USA [44], the UK [45], and China [46] have shown that patients with psoriasis have a higher prevalence of smoking compared to the general population. Moreover, smoking is associated with both the development and disease severity of psoriasis [47]. Thus, patients who smoke may be at risk of more severe forms of psoriasis requiring systemic therapy, which may increase the risk of developing cancer [41,44]. A recent study showed that patients who smoke are at increased risk of developing squamous cell carcinoma. The risk of developing basal cell carcinoma and melanoma cancer appears to not be influenced by smoking [48].

When discussing alcohol as a risk factor, things are less clear. So far, it has not been established that alcohol is a risk factor for developing psoriasis or for developing a more severe form. What has been confirmed is that psoriasis patients consume more alcohol than the general population. Chronic alcohol consumption has been shown to produce immunosuppression, which can lead to a more aggressive disease [40,49,50]. A large prospective study correlated the chronic alcohol consumption of patients with an increased risk of developing skin cancer, especially basal cell carcinoma and squamous cell carcinoma. Analyzing different kinds of alcoholic beverages, the same study found that male patients who predominantly consumed spirits were more likely to develop basal cell carcinoma and melanoma. Patients who drank wine were at risk of developing basal cell carcinoma and squamous cell carcinoma. In female patients who predominantly consumed wine, an increased risk was observed only for basal cell carcinoma. Beer consumption was not associated with an increased risk of developing skin cancer [51].

Thanks to the beneficial effect that sun rays have on the skin, psoriasis patients show sun-seeking behavior. Thus, they spend more time in the sun, are prone to trying home UVB treatments, and some of them even go to tanning salons. This may explain to some extent the higher risk of psoriasis patients developing melanoma or NMSC [24].

Assessing the baseline risk of skin cancer in psoriasis patients is difficult. Patients with severe forms of psoriasis have to take systemic immunosuppressive medication and/or phototherapy, which may in turn increase the risk of cancer. It has been shown that patients who have taken more than 2 years of cyclosporine treatment have a six-fold increased risk of developing NMSC, particularly squamous cell carcinoma [52]. In turn, PUVA treatment is associated with an increased risk of skin cancer even after cessation of the sessions. The increase in risk is directly proportional to the number of therapy sessions. The highest risk was seen in patients who had more than 450 treatment sessions [53]. It has also been noted that methotrexate taken for longer than 2 years is an independent risk factor for squamous cell carcinoma [54].

## 4. Treatment-Related Cancer Risk

Because psoriasis is a chronic disease, most patients have to take therapy for an indefinite amount of time. Therapeutic options for psoriasis patients are numerous. The therapy a psoriasis patient will take is based mostly on the severity of the disease and present comorbidities. Thus, patients with mild forms are generally treated with topical medication, while patients with severe forms are treated with systemic therapy. As mentioned above, the choice of medication is also made considering the patient’s possible comorbidities, as certain medications used in psoriasis may decompensate other comorbidities of the patient [55,56].

The next section will review the main therapeutic options for psoriasis and the possible associated risk for the development of skin cancer.

### 4.1. Topical and Oral Corticosteroids

Topical corticosteroid treatment still maintains its position as the most commonly used form of medication for psoriasis patients. This can be used alone or in combination with a synthetic vitamin D analog (calcipotriol) [7]. A study conducted on mice in Denmark, whose primary goal was to assess the carcinogenic risk of topical therapies, showed that none of the topical treatments used increased the risk of photocarcinogenesis. The topical drugs used in the study were hydrocortisone 17-butyrate, clobetasol 17-propionate, and calcipotriol. Unfortunately, the literature has not addressed this topic much [57].

When it comes to oral glucocorticoids, the results of studies are mixed. Some studies support an association between oral glucocorticoids and the risk of basal cell carcinoma, which seems to increase with the duration and dose of corticosteroids [58]; other studies have found an association between oral glucocorticoids and squamous cell carcinoma [59]. On the other hand, some studies contradict the above hypothesis, especially when discussing young patients and the risk of basal cell carcinoma. Even among patients previously exposed to ultraviolet radiation, the risk could not be confirmed [60,61]. Further studies regarding this topic need to be performed.

### 4.2. Phototherapy

Oral psoralen and ultraviolet A light therapy (PUVA) is very effective in the management of psoriasis. This treatment slows down the division of epidermal cells, which leads to prolonged epidermal turnover and improved symptoms. Nevertheless, at the same time, mutations in the p53 protein may occur, which may contribute to the risk of developing NMSC [62].

A 30-year-old prospective study conducted in the US demonstrated a causal relationship between PUVA and NMSC, particularly squamous cell carcinoma [31]. Patients showed an increased risk even after discontinuation of PUVA cycles and, more interestingly, patients developed skin cancers in areas not exposed to UVA. The risk of developing squamous cell carcinoma was found to be directly proportional to the number of therapy sessions. Patients who had between 351 and 450 sessions had a six-fold higher risk of developing squamous cell carcinoma than patients who had less than 50 PUVA sessions. Patients who underwent less than 150 sessions of therapy were associated with a modest risk of developing squamous cell carcinoma. Bath PUVA does not appear to be associated with a risk of developing NMSC. It is important to mention that the risk of basal cell carcinoma does not increase directly proportionally to the number of therapy sessions. Even among patients exposed to more than 350 sessions, the risk is not at the same level as for squamous cell carcinoma. Interestingly, the results reported in the US study were not reproducible in studies conducted in the US. Thus, the risk of developing NMSC was lower in the EU studies than in the US study [31,53]. Our data are concordant with those above mentioned.

More recent studies conducted in the US claim that patients treated with PUVA have an increased risk of developing melanoma. It has been shown that the risk of developing melanoma occurs more than 15 years after the first exposure to PUVA, and the highest risk is seen in patients who have had at least 250 PUVA sessions. In the most recent follow-up study on PUVA, it was shown that patients who have been exposed to at least 200 therapeutic sessions have a 2.9-fold increased risk of developing melanoma, even after adjustment for age and gender. Another interesting aspect is that European cohorts did not find an increased risk for melanoma [63,64].

UV-B therapy can be divided into broad-band or narrow-band radiation therapy, the latter being the newer technique. Narrow-band UV-B therapy emits near-monochromatic radiation at 311 nm. UV-B therapy is considered a safer therapeutic option than PUVA when assessing the risk of NMSC [36]. Multiple studies investigating the causal relationship between UV-B therapy and NMSC have found no correlation between the two [65,66,67]. The attributable risk per session of UV-B was found to be seven times lower compared to PUVA [36]; however, patients exposed to more than 300 sessions of UV-B showed a modest but significantly increased risk of squamous cell carcinoma and basal cell carcinoma. Furthermore, patients who had previously been treated with less than 100 PUVA sessions and who were subsequently treated with more than 300 UV-B sessions had a higher risk of developing squamous cell carcinoma and basal cell carcinoma in the areas exposed to UV-B radiation. Overall, narrow-band UV-B therapy is a very effective therapy for psoriasis patients and is considerably safer when not combined with PUVA [39,68].

### 4.3. Conventional Systemic Therapies

The most widely used conventional immunosuppressant is methotrexate [39]. It prevents DNA synthesis and thus reduces cell turnover. The doses of methotrexate used for psoriasis patients are typically less than 25 mg per week, which has an immunosuppressive effect. This can put patients at risk for various neoplasms [69]. However, the relationship between methotrexate and the risk of skin cancer is not fully proven [7]. When methotrexate is given in combination with PUVA, the patient has a two-fold risk of developing squamous cell carcinoma [70]. At the same time, methotrexate used in combination with cyclosporine has been found to increase the risk of NMSC [71]. Methotrexate used for more than 2 years becomes an independent risk factor for the development of squamous cell carcinoma [53]. This finding was also made in a 2020 study published by the American College of Rheumatology [72]. The risk of methotrexate for melanoma is controversial. Some studies claim there is a small but significant risk [73,74], while a case–control study in Sweden claims that methotrexate has not been shown to increase the risk of melanoma [32]. The PSOLAR (Psoriasis Longitudinal Assessment and Registry) observational study comes with an interesting finding that methotrexate- or TNF-inhibitor-treated patients have an increased risk of developing basal cell carcinoma, but no effect was seen on squamous cell carcinoma [75]. Methotrexate should be used with caution in patients with diabetes, obesity, or chronic alcohol consumption [76].

Cyclosporine is another immunosuppressive drug used to treat severe forms of psoriasis. The doses of cyclosporine used for psoriasis patients are low compared to those used for organ transplant patients. Intermittent courses of treatment are usually preferred [77]. A prospective study has shown that cyclosporine given for more than 2 years is an important risk factor for developing NMSC. The same study noted that cyclosporine is particularly prone to squamous cell carcinoma development. All patients who were diagnosed with NMSC had received PUVA treatment in the past, and it is most likely that the risk of NMSC was mutually enhanced [52]. In another study, the risk of squamous cell carcinoma development among PUVA-treated patients who also received cyclosporine was three times higher than among psoriasis patients who had never received cyclosporine. It should be mentioned that the highest risk was among patients who had received at least 200 sessions of PUVA in the past [78]. A study conducted in Finland that followed the effects of short-term cyclosporine monotherapy did not find an association between cyclosporine and the risk of squamous cell carcinoma. The median treatment time was 8 months [79].

Acitretin is the most widely used oral retinoid medication. It has both anti-inflammatory and immunosuppressive effects. The combination of PUVA and acitretin appears to be more effective than using either drug alone. Moreover, it appears that the combination of the two drugs reduces the toxicity of PUVA and thus reduces the risk of squamous cell carcinoma [77,80].

Oral fumaric esters are used for the treatment of severe forms of psoriasis mainly in German-speaking countries and northern Europe [77]. The use of oral fumaric esters does not seem to influence the risk of malignancy in patients with psoriasis [7]. Two studies from Germany found no association between NMSC and oral fumaric esters [81,82]. However, there is one study that raises the suspicion of a possible risk of melanoma development among patients treated with oral fumaric acid esters [83].

### 4.4. Biologic Therapies

Biologic therapy is increasingly used to treat severe forms of psoriasis. There are currently four categories of US Food and Drug Administration (FDA)-approved biological drugs for use in psoriasis as follows: tumor necrosis factor inhibitors—TNFi (etanercept, infliximab, adalimumab, and certolizumab); interleukin IL-12/23 antagonists (ustekinumab); IL-17A inhibitors and IL-17 receptor antagonists; and anti-IL-23 agents [84]. Although biological therapies target only the part of the immune system that is overacting, concerns about the possible risk of cancer are justified. The risk of NMSC and melanoma following the use of TNFi has been extensively studied among patients with rheumatoid arthritis and Crohn’s disease, but less so among patients with psoriasis [39].

In patients with psoriasis, the results of studies on the risk of NMSC are heterogeneous [39]. A study of over 42,000 psoriasis patients found no increased incidence of NMSC among patients treated with biologic therapy [85]. Another study in the UK, which studied the risk of NMSC in biological-therapy-treated patients compared to conventional systemic-therapy-treated patients, found no increased incidence of NMSC in patients treated with biological therapy [86]. A meta-analysis including studies from Israel, Italy, Spain, the UK, and Ireland found no correlation between the use of biological therapies and the risk of NMSC [87]. A systematic review found that TNFi, specifically Adalimumab and Etanercept, increases the risk of NMSC, especially squamous cell carcinoma [88]. When comparing the risk of NMSC in TNFi-treated psoriasis patients versus rheumatoid arthritis patients, a higher risk is observed in the psoriasis group. This may be due to past exposure to cyclosporine or PUVA [89]. In the PSOLAR observational study, it was found that TNFi-treated patients had an increased risk of developing basal cell carcinoma, yet not squamous cell carcinoma (Table 1 and Table 2) [75].

The main limitation that may interfere with the results obtained in the studies so far is the long latency period between exposure to a biological drug and the occurrence of an NMSC. Because the biologic therapies used in psoriasis are relatively new, and because of the relatively short post-administration follow-up periods, accurate risk assessment is difficult. More prospective studies with longer follow-up periods are emerging [90].

Studies assessing the risk of developing melanoma among TNFi-treated patients are few and have heterogeneous results. A study that included biological registries of 11 European countries found no increased risk of developing melanoma [91]. Another study comparing the risk of developing cancer between patients treated with TNFi and those on conventional systemic therapies found no increased risk for melanoma [92]. The same conclusion was reached by a study conducted in Germany that investigated the long-term safety of systemic therapies administered in psoriasis [81]. However, there have been studies in rheumatoid arthritis patients in particular that have assessed the risk of melanoma in patients treated with TNFi and found a causal relationship between the two [93,94]. Adalimumab appears to be associated with a higher risk of melanoma [95].

Ustekinumab is an IL12/IL23 antagonist approved for the treatment of psoriasis [7]. Because of its mechanism of action, blocking both the IL12 and IL23 pathways, suspicion has been raised as to whether this class of biological drugs may have a procarcinogenic effect of itself. In mouse studies, this risk was proven; however, in clinical trials, this hypothesis has not been confirmed and no increased incidence of any cancer, including NMSC, has been found [36]. In a study evaluating the long-term safety of ustekinumab, no increased incidence of NMSC or melanoma was observed compared to the general population [95,96]. Another study also evaluated the long-term safety of ustekinumab and observed that the majority of neoplasms occurring were NMSCs, with basal cell carcinoma predominating, but with an incidence comparable to the placebo-treated group [97].

There is currently too little real-world evidence on anti-IL17 and IL23 to assess skin cancer risk [7].

## 5. Conclusions

Psoriasis is a systemic condition that has a strong impact on the patient’s quality of life. Patients with psoriasis are automatically at intrinsic risk of developing neoplasia through chronic inflammation and immune system dysfunction. This risk is increased in patients with severe forms because they require systemic therapy, which in turn may increase the risk of neoplasia. Long-term PUVA therapy has been proven to raise the risk of NMSC, namely squamous cell carcinoma and melanoma, with the risk rising further when paired with immunosuppressive medication. The most researched combination, which increases the risk of NMSC, is PUVA with cyclosporine.

Biologic therapies are becoming more prevalent as an alternative to conventional treatments. Although the risk of malignancy associated with this novel therapy is still unknown, they appear to carry a small risk of NMSC development. It is difficult to assess the risk of malignancy because patients receiving these new therapies have a history of exposure to PUVA and/or immunosuppressive medication. Studies that have analyzed the carcinogenic risk of these drugs have shown heterogeneous results, short follow-up periods, and multiple confounding factors.

Additionally, it is important to emphasize the patient’s risky behaviors, which include drinking more alcohol, smoking more, and spending more time in the sun. These behaviors all raise the risk of non-melanoma skin cancer.

Prospective studies with few confounders and long follow-up periods should be conducted to better understand the factors that lead to the increased risk of skin cancer that patients with psoriasis have.

## Figures and Tables

**Table 1 cancers-15-02451-t001:** TNF-α roles in the regulation of tumor promotion and propagation [84].

TNF-α Roles
Autocrine growth and survival factor for malignant cells
Tissue remodeling by matrix metalloproteinases
Control of leukocyte infiltration by cytokine modulation
Increased tumor motility (invasiveness)
Epithelial–mesenchymal transition
Induction of angiogenic factors
Loss of androgen response
Resistance to cytotoxic medication

**Table 2 cancers-15-02451-t002:** TNF-α roles in combating neoplasia [84].

TNF-α Roles
Blood flow stops, intratumoral bleeding, vascular necrosis (intratumoral injection)
Tumor colonization with PMN
Inhibition of tumor growth by macrophages and NK cells
NK- and LAK-induced destruction and tumor rejection
Antitumor immunity and tumor-induced CTL removal
Favoring cytochrome c mitochondrial-induced apoptosis
c-myc-dependent apoptosis
Inhibition of NF-kB activation

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
