# Peer review of "Skin Cancer Correlations in Psoriatic Patients"

_cancers, 2023, doi:10.3390/cancers15092451_

Round 1

Reviewer 1 Report

This study presents a review of the relationship between skin cancer and psoriasis. They reviewed the available studies that evaluate the risk of skin cancer in psoriatic patients. This is a well-organized review of recent studies to show the current perspectives about this theme. One minor issue that I want to point out is that there were some repetitions of similar sentences in few parts.

Author Response

Thank you so much for your appreciations and for your work in reviewing our manuscript.

Reviewer 2 Report

In the present manuscript, authors have explored "Skin cancer correlations in psoriatic patients". The subject is of interest and falls in the topics of “Cancers” Journal. The rticle is acceptable with minor corrections.

After reviewing the manuscript thoroughly, I have following comments:

What is rationale of skin cancer correlations in psoriatic patients?

Abstract must be improved.

Check the abbreviations throughout the manuscript and insert the where required.

The English language needs to improve throughout the manuscript.

Author Response

What is rationale of skin cancer correlations in psoriatic patients?

The rationale behind our manuscript is both a theoretical and a practical one. On the practical level, we are involved in the treatment of psoriatic patients for more than 20 years and observed that it seems to be an increased prevalence of skin neoplasia among patients both in those treated with classical and modern drugs. On a theoretical level, we are studying the relationships between inflammation, psoriasis, skin cancers, and modern therapies such as biologicals and small molecules.

Abstract must be improved.

Unfortunately, we were constrained by the maximum 200 words limit.

Check the abbreviations throughout the manuscript and insert the where required.

Done.

The English language needs to improve throughout the manuscript.

Thank you for the recommendation. Concerning the English language we used Grammarly, and it suggests that now there are no more language issues.

Reviewer 3 Report

This review is focused to establish correlation between psoriasis treatment and intricate risk of skin cancer in psoriatic patients. The manuscript addresses the issue in focused way. The following points may also be addressed.

1.       Use of Calcineurin inhibitors for the psoriasis and its relation with skin cancer

2.       Pathway by which PUVA increase the risk of melanoma may be explained.

3.       Effect of use of common topical creams may be addressed.  

Author Response

Use of Calcineurin inhibitors for the psoriasis and its relation with skin cancer

It is a very natural and pertinent comment. The reason we did not address this issue is (and that is also the reply to your third comment) that we are preparing a manuscript that will discuss the relationships between topical treatments in inflammatory skin diseases and their relation with skin cancers. For that reason, we even considered initially excluding the few ideas concerning topical steroid treatments, but because of their daily presence in the psoriasis treatment, we decided to mention them.

Pathway by which PUVA increase the risk of melanoma may be explained.

We considered that those pathways are well known and describing them will not major influence the presentation of our theme. Yet, if your opinion is that presenting them would be an asset for the article, please let us know and we will do that.
Anyway, thank you so much for your work and for your valuable comments.

Reviewer 4 Report

This article is interesting in its approach to skin cancer and psoriatic disease.

I believe that there are many more references than necessary and some could be deleted.

The English needs to be revised in some points.

Author Response

Thank you so much for your hard work in reviewing our manuscript.

Concerning the English language we used Grammarly, and it suggests that now there are no more language issues.